# A Functional ClpXP Protease is Required for Induction of the Accessory Toxin Genes, *tst, sed,* and *sec*

**DOI:** 10.3390/toxins12090553

**Published:** 2020-08-28

**Authors:** Jenny Schelin, Marianne Thorup Cohn, Barbro Frisk, Dorte Frees

**Affiliations:** 1Division of Applied Microbiology, Department of Chemistry, Lund University, SE-221 00 Lund, Sweden; jenny.schelin@tmb.lth.se (J.S.); barbro.k.frisk@gmail.com (B.F.); 2Department of Veterinary and Animal Sciences, University of Copenhagen, 1870 Frederikberg C, Denmark; MTUC@novozymes.com

**Keywords:** staphylococcus aureus, TSST-1, staphylococcal enterotoxins, virulence gene regulation, ClpXP, mobile genetic elements, RNA-seq

## Abstract

Staphylococcal toxic shock syndrome is a potentially lethal illness attributed to superantigens produced by *Staphylococcus aureus*, in particular toxic shock syndrome toxin 1 (TSST-1), but staphylococcal enterotoxins (SEs) are also implicated. The genes encoding these important toxins are carried on mobile genetic elements, and the regulatory networks controlling expression of these toxins remain relatively unexplored. We show here that the highly conserved ClpXP protease stimulates transcription of *tst* (TSST-1), *sec* (SEC), and *sed* (SED) genes in the prototypical strains, SA564 and RN4282. In the wild-type cells, the post-exponential upregulation of toxin gene transcription was proposed to occur via RNAIII-mediated downregulation of the Rot repressor. Contradictive to this model, we showed that the post-exponential induction of *tst*, *sed,* and *sec* transcription did not occur in cells devoid of ClpXP activity, despite the Rot level being diminished. To identify transcriptional regulators with a changed expression in cells devoid of ClpXP activity, RNA sequencing was performed. The RNAseq analysis revealed a number of global virulence regulators that might act downstream of ClpXP, to control expression of *tst* and other virulence genes. Collectively, the results extend our understanding of the complex transcriptional regulation of the *tst, sed,* and *sec* genes.

## 1. Introduction

*Staphylococcus aureus* colonizes the anterior nares and the skin of approximately 30% of the human population. Colonization is associated with increased risk of infections, ranging in severity from relatively minor skin and soft tissue infections, to life-threatening bacteremia and toxic shock syndrome (TSS) [1]. While most *S. aureus* infections involve the coordinated expression of numerous virulence factors, a few select diseases, including TSS, are mediated by single staphylococcal exotoxins, belonging to the family of *S. aureus* superantigens (SAgs) [2,3]. These SAg exotoxins lead to massive activation of T cells, and symptoms associated with TSS include, fever, skin rash, desquamation, hypotension, and hemodynamic shock [4,5]. TSS can occur in healthy menstruating women using intravaginal protection, such as tampons, and is colonized by *S. aureus* carrying the *tst* gene, encoding the toxic shock syndrome toxin 1 (TSST-1) [3]. This toxin is also the causative agent of approximately half of all surgical-related TSS cases [3]. The remaining cases of TSS are triggered by enterotoxins belonging to the SAg family of toxins [3]. These staphylococcal enterotoxins (SEs) are additionally associated with staphylococcal food poisoning, a foodborne intoxication caused by the consumption of preformed SEs in foods, resulting in vomiting and diarrhea [6]. Of notice, the emetic activity of SEs seems unrelated to their superantigenicity [3,4]. The two most common SEs associated with food poisoning are staphylococcal enterotoxins A (SEA) and D (SED) [6,7,8,9]. Almost all *S. aureus* SAg exotoxins are encoded on mobile genetic elements that can be exchanged by horizontal gene transfer [3]. As an example, the *sed* gene (encoding SED) is present on a plasmid [10], while the *tst* gene is carried by *S. aureus* pathogenicity islands (SaPIs) [11], phage-like elements that are unable to mobilize by themselves, but require specific helper phages for packaging into phage capsid and for dissemination [12].

To coordinate expression of virulence factors, *S. aureus* strains make use of complex networks of transcriptional regulators [13]. The most thoroughly investigated virulence regulator is the Agr quorum sensing system that induces expression of RNAIII (a regulatory RNA molecule controlling virulence gene expression) at high cell density, by impacting the stability and translation of several mRNAs [14,15,16]. The Agr system positively controls transcription of *tst* and select enterotoxin genes, like *sed* [12,17,18,19,20,21]. This *agr-*dependent upregulation of *tst* and *sed* transcription was proposed to be achieved via RNAIII-mediated downregulation of Rot (Repressor of toxin), thereby relieving Rot repression of the *sed* and *tst* genes [22,23]. Additionally, a large number of two-component systems, the alternative sigma factor, SigB, the ClpXP protease, and a number of transcriptional regulators, belonging to the family of SarA homologs, contribute to control *S. aureus* virulence genes expression [13,24,25,26,27]. The paradigm of *S. aureus* virulence regulation was, however, established in a limited number of model strains that did not harbor accessory toxin genes, such as the *tst* gene, and other SE genes. Hence, the regulatory networks controlling transcriptional regulation of *tst* and SE genes remain relatively unexplored.

The highly conserved cytoplasmic ClpXP protease is required for the growth-phase dependent induction of the *hla* gene (α-toxin) and numerous other virulence genes encoded by the core genome of *S. aureus* strains [26,28,29,30]. Here, we asked if the ClpXP protease controls the expression of SAg exotoxins in SA564 and RN4282, two *tst* prototypical strains of clinical origin [7,31]. Strikingly, the post-exponential induction of *tst, sed,* and *sec* (encoding SEC) transcription did not take place in the SA564 cells devoid of ClpXP activity, despite the Rot level being diminished below the wild-type level. A reduction in Rot-level, therefore, did not per se lead to transcriptional induction of the accessory toxin genes. To elucidate the pathway through which ClpXP controls *tst* transcription, RNA sequencing was performed in RN4282 wild-type and RN4282 devoid of ClpXP activity. This analysis revealed that the transcriptional regulators SarU, SarX, and MgrA could potentially work downstream of ClpXP to control the expression of genes encoding SAg exotoxins.

## 2. Results and Discussion

### 2.1. Transcription of the Accessory Toxin Genes sed, sec, and tst is Induced in the Post-Exponential Growth Phase in the SA564 Strain Background

To examine the role of ClpXP in transcriptional regulation of toxin genes encoded by mobile genetic elements we chose strain SA564, a CC5/t648 strain recovered from a patient with toxic shock syndrome, as a model [31]. In SA564, the *tst* gene is encoded by a *S. aureus* pathogenicity island that is closely related to SaPIn1 in strain N315 and which also carries the *sec* gene [32,33]. Strain N315 was previously reported to display a low level of *tst* transcription, which was linked to defective RNAIII production in this strain background [34]. To examine the expression of RNAIII and toxin genes in the SA564 background, we performed Northern blot analysis with RNA extracted in late-exponential (OD_600_ = 1.5), early post-exponential growth phase (OD_600_ = 3.0), or late post-exponential growth phase (OD_600_ = 6.0). These analyses demonstrated that transcription of the SaPI encoded *tst* and *sec* genes, similarly to the transcription of RNAIII, was induced when the SA564 cells enter the post-exponential phase (Figure 1). The SA564 strain harbors plasmid pIB485 encoding SED, and the Northern blotting analysis additionally revealed strong upregulation of the *sed* transcription in post-exponential SA564 wild-type cells (Figure 1). Transcription of the *tst* gene responds to metabolic cues, and carbon catabolite protein A (CcpA) acts as a repressor of TSST-1 [34]. Consistent with this finding, we found that transcription of *tst* was reduced upon addition of 10 mM glucose to the growth medium, but only at the latter time-point (OD_600_ = 6.0; Figure 1). In contrast, transcription of *sec* and *sed* was not affected by the addition of glucose at this time-point. Taken together, the presented results showed that the transcriptional regulation of *sec*, *sed*, and *tst* in SA564 followed the paradigm established in other strain backgrounds, making SA564 a good model strain for studying the transcriptional regulation of the accessory toxin genes.

### 2.2. ClpXP is Required for Post-Exponential Induction of sed, sec, and tst Transcription

The ClpXP protease is composed of separately encoded proteolytic subunits (ClpP) and substrate recognition subunits (ClpX) that associate to form the active protease [35]. To examine if ClpXP activity contribute to the transcriptional regulation of *sec, sed*, and *tst,* we analyzed the expression of the three genes in SA564 cells, where either *clpP* or *clpX* were deleted. Interestingly, the post-exponential induction of *sed* and *sec* transcription observed in the SA564 wild-type cells did not take place in SA564Δ*clpX* and SA564Δ*clpP* cells (Figure 1). Similarly, transcription of *tst* did not increase in post-exponential SA564Δ*clpX* cells. In contrast, induction of *tst* transcription was observed in late post-exponential SA564Δ*clpP* cells (OD_600_ = 6.0), grown in the absence of glucose. As in wild-type cells, addition of glucose to the growth medium, prevented the upregulation of *tst* in SA564Δ*clpP* cells. Taken together, the Northern blotting analysis revealed that both ClpX and ClpP are required for the growth-phase-dependent upregulation of *sec* and *sed* transcription, supporting the ClpX and ClpP control transcription of these genes through the formation of the ClpXP protease. In contrast, delayed upregulation of *tst* transcription was observed in the SA564Δ*clpP* cells, but not in the SA564Δ*clpX* cells, indicating that ClpX might control *tst* transcription via its ClpP-independent chaperone activity.

### 2.3. SED and TSST-1 Protein Levels are Reduced in SA564ΔclpX and SA564ΔclpP

We next monitored if the reduced transcription of the toxin genes was reflected in the amount of SED and TSST-1 toxins recovered from the growth medium of cells lacking ClpX or ClpP. This was done by performing quantitative ELISA on the spent supernatant derived from the cultures of SA564 wild-type, SA564Δ*clpX*, and SA564Δ*clpP*, throughout the growth curve (Figure 2). As expected, the normalized concentrations of SED and TSST-1 toxins detected in the growth medium of SA564 wild-type cells were significantly higher (SED *p* < 0.001; TSST-1 *p* = 0.02) at 24 h than at 2 h (Figure 2B,C), which was consistent with the post-exponential induction of *sed* and *tst* transcription in wild-type cells. The SED concentrations produced at 24 h by the SA564Δ*clpX* (*p* < 0.001) and SA564Δ*clpP* (*p* < 0.001) cells were significantly lower than those produced by the wild-type in the post-exponential growth phase, which agreed well with the lack of transcriptional induction of *sed* in these strains. For TSST-1, the picture was rather similar. In SA564Δ*clpX*, the low levels and absence of induction of *tst* transcripts corresponded to the significantly lower (*p* = 0.03) concentration of TSST-1, compared to the SA564 wild-type at 24 h. For SA564Δ*clpP*, the concentration of TSST-1 was again significantly lower (*p* = 0.02) compared to the SA564 wild-type. Hence, the reduced transcription of the *sed* and *tst* genes was reflected in the diminished SED and TSST-1 protein levels in cells devoid of ClpXP activity. However, we noted that there were significant (*p* < 0.001) differences in the growth rates between the wild-type and mutant strains at 37 °C (Figure 2A). The difference was especially pronounced for SA564Δ*clpP*, with the slowest growth rate (μmax 0.57 ± 0.03 h^−1^) and lowest maximum OD at 24 h, compared to the SA564 wild-type (μmax 2.03 ± 0.06 h^−1^) and SA564Δ*clpX* (μmax 1.20 ± 0.14 h^−1^). This difference in growth capacity, and the resultant physiological fitness of the strains, is important to take into consideration when comparing the concentrations of toxins produced during growth, as these are often correlated.

### 2.4. Diminished Levels of the Rot Repressor does not per se Increase Transcription of the sed, sec, and tst Genes

According to one current model, RNAIII upregulates *tst* transcription by blocking translation of r*ot* mRNA, thereby, relieving Rot repression of the *tst* promoter [14,15,23,36]. To investigate if ClpXP controls transcription of toxin genes via the RNAIII/Rot pathway, we determined the levels of RNAIII transcript and Rot protein in SA564Δ*clpX* and SA564Δ*clpP* cells in different growth phases. The Northern blot analysis revealed similar levels of RNAIII in SA564 wild-type and mutant cells at OD_600_ = 1.5 and OD_600_ = 3.0, while the levels of RNAIII appeared to be slightly decreased in the mutants at OD_600_ = 6.0 (Figure 1). Notably, despite the levels of RNAIII not being increased in SA564Δ*clpX* cells, the levels of Rot protein were clearly diminished in the SA564Δ*clpX* cells at all time-points, as was also described for the *clpX* deletion mutants constructed in other strain backgrounds [37]. Despite the low Rot content, there was very low expression and no post-exponential induction of *sed*, *tst*, and *sec* in the SA564Δ*clpX* strain (Figure 3). From this result, we conclude that a reduction in the cellular Rot-level did not per se lead to the induction of transcription of the accessory toxin genes, hinting that other transcriptional regulators contribute to the upregulation of *tst, sec*, and *sed* transcription in post-exponential growth phase.

### 2.5. A Functional ClpXP Protease is Required for TSST-1 Expression in RN4282

To establish if the ClpP and ClpX exerted control of *TSST-1* expression is conserved in other strain backgrounds, the *clpP* and *clpX* genes were inactivated in the prototypic RN4282 strain used for the first characterization of the *SaPI1* and the *tst* gene [11,38]. TSST-1 production in the RN4282 wild-type and *clp* mutants was next determined in the ON cultures (18 h of incubation at 37 °C) of the three strains. Consistent with the results obtained in SA564, inactivation of either *clpP* or *clpX*, significantly (Δ*clpP p < 0.001*; Δ*clpX p < 0.001*) reduced the production of TSST-1, and the reduction was more pronounced in the RN4282Δ*clpP* cells than in the RN4282Δ*clpX* cells (Figure 4). In *S. aureus*, ClpP can associate with an alternative substrate recognition factor, ClpC, to form the ClpCP protease, which is the major protease for removing misfolded and damaged proteins [26]. To rule out that the changes in TSST-1 expression in RN4282Δ*clpP* is associated with the disruption of ClpCP activity, we next examined TSST-1 levels in RN4282 encoding a mutant variant of the ClpX, ClpX_I265E_, which had a single amino acid substitution in the ClpP recognition motif, IGF, of ClpX [26]. Accordingly, cells expressing the ClpX_I265E_ variant, while unable to form the ClpXP protease, retain normal ClpCP activity [26]. As shown above, deletion of *clpP* conferred a severe fitness cost to *S. aureus* that might indirectly impact global gene expression. In contrast, *S. aureus* cells expressing the ClpX_I265E_ variant grow similar to the wild-type cells at the laboratory conditions used here [26]. Importantly, as can be seen in Figure 4, the expression of the ClpX_I265E_ variant eliminated TSST-1 production in RN4282, confirming that a functional ClpXP protease is essential for the ability of S. *aureus* to produce this important toxin.

### 2.6. RNA-seq to Establish Transcriptional Regulators Involved in ClpXP-Mediated Control of tst Transcription in RN4282

The results presented above suggest that the ClpXP-dependent upregulation of *tst* transcription did not involve Rot de-repression. To search for transcriptional regulators potentially working downstream of ClpXP, RNA sequencing was performed to identify virulence regulatory genes exhibiting altered expression, upon inactivation of the ClpXP protease. The RN4282 strain is a preferred model strain for studying transcriptional regulation of the *tst* gene [39,40]. For this reason, the RNA-seq analysis was performed in the RN4282 background, using RNA samples derived from post-exponential cultures (OD_600_ = 3.0 ± 0.1) of RN4282 wild-type, and RN4282 expressing the ClpX_I265E_ variant. The RNA-seq analysis confirmed that the transcription of *tst* is significantly downregulated in RN4282clpX_I265E_ cells (3.1 fold; *p* < 0.001), and so was the expression of the SAPI1 encoded *ear* gene (5.7 fold; *p* < 0.001) (Appendix A). Similar to pro-phages, SaPI elements exhibit a typical modular organization, with two divergent transcription units controlled by a phage-like genetic switch [12]. Expression of the core SaPI1 genes was similar in the RN4282 wild-type and RN4282clpX_I265E_ (Appendix A), signifying that ClpXP controls the transcription of the SaPI1-encoded virulence genes, *ear* and *tst,* without impacting the control of the SaPI regulatory switch. In contrast, inactivation of ClpXP was previously shown to promote excision and replication of SaPI5 in the USA300 strain background, which encoded an Stl repressor, unrelated to the Stl repressor encoded by SaPI1 [26,41].

Previous studies found *tst* transcription in strain RN4282 to be subject to strong negative regulation through the transcriptional regulator SarA, which binds directly to the *tst* promoter, and indirectly by the alternative Sigma factor, SigB [39,40]. We, therefore, first checked if expression of the *sarA* or *sigB* genes was increased in RN4282 expressing the ClpX_I265E_ variant, however, this was not the case (Appendix A). In contrast, transcription of *asp23*, a gene transcribed solely from SigB-dependent promoters [42], was reduced two-fold in RN4282clpX_I265E_ cells (Appendix A). This ruled out that the lowered *tst* transcription was accomplished via enhanced SigB activity. Additionally, we did not observe a significant change in transcription of the *S. aureus* exotoxin expression (Sae) regulatory operon, which in other strain backgrounds is a direct and dominant positive regulator of *tst* transcription [43]. The RNA-seq analysis, however, revealed a significant change in transcription of a number of genes encoding global virulence regulators, such as the SrrAB and AgrACBD, two component systems, and the SarA-like transcriptional regulators SarU, SarX, SarS, MgrA, and SarT (Table 1 and Appendix A). Of these regulators, only Agr and SrrAB were previously confirmed to control the transcription of the *tst* gene [22,23,44,45]. In RN4282clpX_I265E_ cells, transcription of RNAIII and the *agrACBC* operon was reduced 2–2.3-fold, a minor reduction that did not result in a diminished transcription of known RNAIII-controlled genes, such as *hla* (α-hemolysin), or the AgrA-controlled *psmβ1* and *psmβ2* (PSMs) genes—Table 1 and Appendix A [14,46]. Based on this finding, we question if the two-fold reduction in transcription of the *agr* locus caused the downregulation of *tst* transcription in RN4282clpX_I265E_ cells. The SrrAB (staphylococcal respiratory response) two-component-system was found to be a negative regulator of *tst* transcription [44]. Deletion of the *srrAB* genes, however, did not upregulate TSST-1 expression in the *tst^+^* model strain, MN8, indicating that *tst* transcription is not under strong negative regulation by the SrrAB system, under standard laboratory conditions [47]. Consistent with the latter finding, *tst* transcription was reduced in the RN4282clpX_I265E_ cells, even though the expression of the *srrAB* operon was reduced 3-fold (Table 1 and Appendix A). Therefore, the positive effect of ClpXP on *tst* transcription did not seem to be mediated via SrrAB.

Transcription of genes encoding the SarA-homologs SarU, SarX, SarS, and MgrA, was downregulated in the RN4282clpX_I265E_ cells, while transcription of *sarT* was reduced (Table 1). Thus, one plausible hypothesis is that ClpXP controls *tst* transcription via pathways involving one or more of these SarA-homologs. The most downregulated virulence regulator gene in RN4282clpX_I265E_ cells, *sarU*, encodes a relatively uncharacterized transcriptional regulator that was, however, not expressed in 6 out of 7 *S. aureus* model strains examined (JE2, SH1000, MW2, Newman, COL, and UAMS-1) [48]. For this reason, we propose that the conserved role of ClpXP in *S. aureus* virulence regulation is not mediated via SarU. Strikingly, deletion of the *clpP* gene, reduced the level of the MgrA transcriptional regulator in 5 out of 5 examined *S. aureus* strains [24]. Here, we observed a 2-fold reduction in the level of *mgrA* transcript in cells expressing the ClpX_I265E_ variant. Taken together, these findings suggest that ClpP has a conserved positive effect on MgrA expression, which is mediated via the formation of the ClpXP protease. MgrA directly stimulates the transcription of toxin genes encoded by the core genome [49]. Hence, the conserved positive effect of ClpXP on *S. aureus* virulence gene expression might be accomplished by upregulation of MgrA. The role of MgrA in transcriptional control of *tst, sed*, and other enterotoxin genes has, to our knowledge, not been explored, as the MgrA regulon was established in *S. aureus* strains not carrying these genes [50,51].

## 3. Concluding Remarks

Production of the SAg exotoxins causes TSS, one of the most severe disease manifestations associated with *S. aureus* colonization. The amounts of SAg exotoxins produced by clinical *S. aureus* isolates vary greatly [52]. The variation in toxin expression seems to not be accomplished by the Agr system [52], and the regulatory systems responsible for this variation remain largely unknown. Here, we showed that a functional ClpXP protease is required for transcriptional induction of the *sec, sed*, and *tst* genes in prototypical strains carrying these genes. Our results suggest that the role of ClpXP in transcriptional upregulation of these accessory toxin genes is not mediated via RNAIII/Rot, or other known regulators of these genes. Instead, global transcriptional analysis suggested that ClpXP controls expression of the *tst* gene via one or more of the transcriptional regulators SarU, SarX, SarT, and MgrA. Importantly, the role of these regulators in virulence gene expression was established in model strains not harboring the *tst, sec*, or *sed* toxin genes, and therefore, their role in the transcriptional regulation of these accessory toxin genes remain unexplored [50,51,53,54,55]. Collectively, the presented results extend our understanding of the complex transcriptional regulation of the *tst, sed*, and *sec* genes, and emphasize the importance of studying virulence regulation in model strains carrying the mobile genetic elements encoding this important class of virulence genes.

## 4. Materials and Methods

### 4.1. Bacterial Strains

Two *S. aureus* wild-type strains were used in this study—SA564 is a CC5/t648 strain isolated from a patient with toxic shock syndrome in USA in 2007, while RN4282 was the prototypic strain used for the first characterization of the *tst* genes [11,31]. Isogenic, SA564Δ*clpX*, and SA564Δ*clpP* mutants were constructed by allelic replacement [30,37]. Whole genome sequencing confirmed that the desired in-frame deletion in *clpX* was the only genetic difference between the SA564 wild-type and SA564Δ*clpX* [56]. RN4282Δ*clpP*, RN4282Δ*clpX*, and RN4282clpX_I265E_ were constructed by generalized transduction, using phage 80α, as described previously [31]. Strains were stored as glycerol stocks at −80 °C and were resuscitated by streaking on Tryptic Soy agar, TSA (Difco Laboratories, BD Diagnostic System, Pont-de-Claix, France) or TSA supplemented with erythromycin (10 μg × mL^−1^, Sigma-Aldrich, Stockholm, Sweden).

### 4.2. Growth Curve Analyses of the SA564 Strains and ON Cultures of the RN4282 Strains

SA564 wt strain, isogenic SA564Δ*clpX* and SA564Δ*clpP* mutants were plated from frozen stocks (−80 °C) onto tryptic soy agar (TSA) (Difco Laboratories, BD Diagnostic System, Pont-de-Claix, France) and grown aerobically overnight (O/N) at 37 °C. Pre-experiment cultures were then prepared by inoculation of 50 mL tryptic soy broth (TSB) (Difco Laboratories, BD Diagnostic System, Pont-de-Claix, France) in baffled E-flasks and grown aerobically O/N (16–18 h) at 37 °C. The experimental cultures for growth curve measurements were started at an initial OD620 of ~ 0.1, at time-point 0, using the O/N pre-experiment cultures. All growth experiment cultures were incubated in a 37 °C water bath, rotation ~160 rpm. To avoid the carry-over of enterotoxins produced in the pre-experiment cultures, the cells were washed prior to inoculation. The cells were centrifuged in a swing-out rotor at 3220× *g* (Eppendorf Centrifuge 5810 R, Hamburg, Germany) for 5 min at 4 °C, the supernatant was removed and the cells were dissolved in 10 + 20 mL of 0.9% sterile NaCl (Merck, Darmstadt, Germany). The cells were centrifuged as above, the supernatant removed, and the cells re-dissolved in TSB. Samples were collected for growth, RNA, SED, and TSST toxin analyses at the following time points—0, 1–6, 24, 48, 72, and 144 h. For each growth curve and strain, three biological replicates were made. ON cultures for the RN4282 wild-type, RN4282Δ*clpP*, RN4282Δ*clpX*, and RN4282clpX_I265E_ strains were prepared and cultivated in the same way as the pre-experiment cultures of SA564 strains. For each strain, growth and toxin samples from one biological replicate (*n* = 1) were collected and analyzed in three technical replicates.

### 4.3. Northern Blot Analysis

To inoculate cultures, a streak of small single colonies was transferred to 25 mL TS broth in a 250 mL baffled Erlenmeyer flask and incubated at 37 °C, 200 rpm (the starting OD was always below 0.1). The RNA extraction and Northern blotting were performed as described previously [37], except that *S. aureus* cells were harvested at a late exponential growth phase (OD600 = 1.5 ± 0.1), at an early post-exponential phase (OD600 = 3.0 ± 0.1), or a late post-exponential phase (OD600 = 6.0 ± 0.1). Cells were quickly cooled on an EtOH/dry ice bath and frozen at −80 °C, until extraction of RNA. In brief, the cells were lysed mechanically using the FastPrep system (Bio101; Q-biogene, Cedex France), and RNA was isolated using the RNeasy mini kit (Qiagen, Hilden, Germany), according to the manufacturer’s instructions. Total RNA was quantified and checked for quality, using the nano-drop2000, and 5 μg of RNA from each preparation was loaded onto a 1% agarose gel. The hybridization was performed, as described in [30], using gene-specific probes labeled with ^32^PdCTP (Perkin-Elmer, Skovlunde, Denmark), using the Ready-to-Go DNA-labeling beads from GE-healthcare. Internal fragments used as templates in the labeling reactions were obtained by PCR, using the primers: *tst* (probe length 559 bp): tst-f: 5′- GCTTGCGACAACTGCTACAT + tst-r: TGGATCCGTCATTCATTGTTAT; RNAIII (probe length 352 bp) (5′- GATCACAGAGATGTTATGG + 5′-CATAGCACTGAGTCCAAGG); *sed* (probe length 449 bp): SED1-forward: 5′-CTATGGTGGTAATATCTCCT [57] + GSEDR2: 5′-ATTGGTATTTTTTTTTCTTTC [58]), and *sec* (probe length 615 bp): sec1-forward 5′- CGCCTGGTGCAGGCATCATA + sec1-reverse 5′- AGCAGAGAGTCAACCAGACCCT. In order to use the same RNA-membrane for several rounds of hybridization, the membranes were stored dry in dark for approximately 8 weeks, to allow the 32P-signal to faint before re-hybridization.

### 4.4. Western Blotting

Cells for isolation of total proteins were harvested at the same time-points and from the same cultures as that used for RNA preparation (Northern blot analysis described above). One milliliter aliquots were harvested and kept at −80 °C, until all samples were collected. The cell pellets were thawed on ice, suspended in 50 mM TrisHCl, pH = 7.4, to a calculated OD_600_ of 10.0. PMSF, Dnase, Rnase, and lysostaphin were added to the samples, and they were incubated at room temperature for 15 min. Cellular debris was removed by centrifugation and the protein concentration of each sample was measured using the Bradford dye-binding procedure from Bio-Rad. For immunoblotting, a total of 5 μg of each sample was loaded on NuPAGE^®^ 4–12% Bis-Tris gels (Invitrogen), using MOPS-Buffer (Invitrogen). The proteins were blotted onto a polyvinylidene difluoride membrane, using the XCell II blotting module (Invitrogen). The membranes were pre-blocked with IgG, before probing with antibodies directed against the Rot transcriptional regulator (Rabbit-anti-Rot-antibody [37]) at a 1:2000 dilution. Bound antibody was detected with the WesternBreeze Chemiluminescent Anti-Rabbit kit or Anti-mouse kit (Invitrogen). The Western blot analysis was repeated three times with similar results.

### 4.5. ELISA

ELISA analysis was performed according to the revised laboratory protocol originally designed for the detection of SEA, as previously described by Wallin-Carlquist et al. [59]. In this study, affinity-purified antibodies (IgG), sheep anti SED, or TSST coating IgG, and biotinylated sheep anti SED or TSST detection IgG (Toxin Technology Inc., Sarasota, FL, USA) were used. Absorbance was measured at 405 nm with Multiskan Ascent^®^ spectrophotometer (Electron Corporation, Thermo Fisher Scientific, Waltham, MA, USA). For standard curves, highly purified SED or TSST toxins (Toxin Technology Inc., Sarasota, FL, USA) were prepared in TSB in a 2X dilution series from 10 to 0.039 ng/mL, and analyzed in triplicate technical replicates. Obtained mean absorbance values for the standard samples were plotted against the known concentrations of SED and TSST and standard curves were created. When using an IgG coating concentration of 2 μg/mL, the linear detection window (LOD) according to the average standard curve (*n* = 3) was between 0.31 to 5 ng/mL. The concentrations of unknown samples were determined using the linear regression, expressed in ng × mL^−1^ of toxin and normalized to the corresponding OD_620_ nm value. For SED, samples from three biological replicates (*n* = 3) were analyzed and for TSST, samples from two biological replicates (*n* = 2) were analyzed. SED and TSST normalized concentrations data of the wild-type and the respective mutants were confirmed to be normally distributed and were compared using Student’s *t*-test, with a two-tailed distribution and equal variance in Microsoft Office Excel (2013). Differences were considered to be significant at *p* < 0.05.

### 4.6. RNA Sequencing

#### 4.6.1. Extraction and Library Preparation and RNA Sequencing

RNA was isolated from two biological replicates grown on different days—25 mL TSB in a 250 mL Erlenmeyer flask were inoculated with RN4282 wild-type or RN4282clpX_I265E_ to a starting OD_600_ below 0.1, and grown at 37 °C with vigorous shaking. When the cultures reached OD_600_ = 3.0 ± 0.1, the samples were withdrawn for the isolation of RNA, as described above. High quality RNA was delivered to DNASense ApS (Ålborg, Denmark) for transcriptomic analysis. Here, sample RNA concentrations were measured in duplicate, using the Qubit BR RNA assay, and the RNA quality was validated for the selected samples using TapeStation with the RNA ScreenTape (Agilent Technologies). All four samples were rRNA-depleted using the Ribo-zero Magnetic kit (Illumina, San Diego, CA, USA), and residual DNA from RNA extraction was removed using the DNase MAX kit (MoBio Laboratories Inc, Qiagen, Hilden, Germany). The samples were purified using the standard protocol for CleanPCR SPRI beads (CleanNA, Waddinxveen, NL) and further prepared for sequencing, using the NEBNext Ultra II Directional RNA library preparation kit (New England Biolabs, Ipswich, MA, USA). Library concentrations were measured using the Qubit HS DNA assay and the library size was estimated using TapeStation with D1000 ScreenTape. The 4 samples were pooled in equimolar concentrations and sequenced (2 × 150 bp) on a HiSEQ X platform (Illumina, San Diego, CA, USA). All kits were used as per the manufacturer’s instructions.

#### 4.6.2. Analysis of Gene Expression, Bioinformatic Processing and Analysis

Scaffolds from the RN4282 genome were annotated using prokka (v.1.14.6) and the pan-genome of all genes from the 5 model strains of *S. aureus* are available from AureoWiki—AureoWiki accessions: COL (*NCBI 2017-03-02*), N315 (*NCBI 2017-03-02*), NCTC8325 (*NCBI, PMID 27035918 UniProt 2016-08-03*), Newman (*NCBI 2017-03-02*), and USA300_FPR3757 (*NCBI 2017-03-02*)) [60]. The forward raw sequence reads in the fastq format were trimmed using USEARCH (v11.0.667) and matched against the prokka-annotated RN4282 reference genome [61]. Reads from the PhiX spike-in were filtered from each sample using USEARCH—filterphix before the output reads filtered based on Q-scores using USEARCH—fastqfilter with max expected errors set to 1.0 (-fastq_maxee 1.0). All reads passing the quality filters were truncated to 150 bp with the option-fastq_trunclen. The trimmed transcriptome sequences were then mapped to the annotated RN4282 reference genome using BowTie2 with the very sensitive option enabled [62]. The output .sam files from BowTie2 were then sorted by sequence gene identifier and converted to .bam files using samtools, and the count-tables were subsequently made using featureCounts (v2.0.1) [63]. Where nothing else was stated, the default settings were used for all tools. The count-tables were imported to RStudio [64] and processed using the default DESeq2 workflow and visualized using ggplot2 [65].

## Figures and Tables

**Figure 1 toxins-12-00553-f001:**
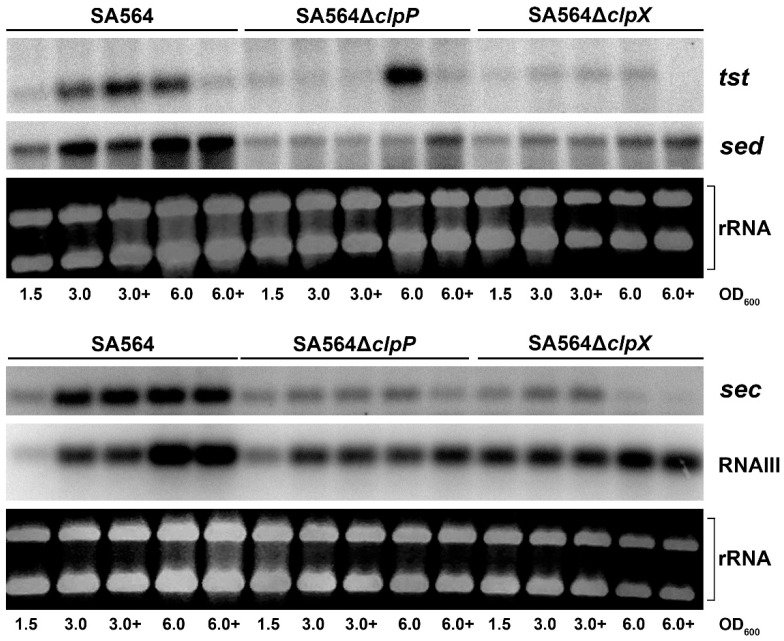
ClpXP is required for post-exponential induction of *sed, sec*, and *tst* transcription. The levels of *tst, sec, sed*, and RNAIII transcript were visualized by Northern blot analysis—at OD_600_ = 1.7, each of the three cultures were split in two, and 10 mM glucose was added to one of the cultures (3.0+ and 6.0+). Cells were harvested for RNA extraction in the late-exponential (OD_600_ = 1.5), early post-exponential growth phase (OD_600_ = 3.0), or the late post-exponential growth phase (OD_600_ = 6.0), and 5 μg of RNA was loaded in each lane of an agarose gel (equal loading was confirmed by visualization of the ribosomal RNAs) and blotted to a Nylon-membrane before hybridizing to radioactively labeled probes, as described in the experimental section. The analyses were repeated twice with similar results. In the depicted pictures, the ^32^P-labeled tst and sed probes were hybridized to the same membrane, while another membrane was used for hybridization to the ^32^P-labeled RNAIII and sec probes.

**Figure 2 toxins-12-00553-f002:**
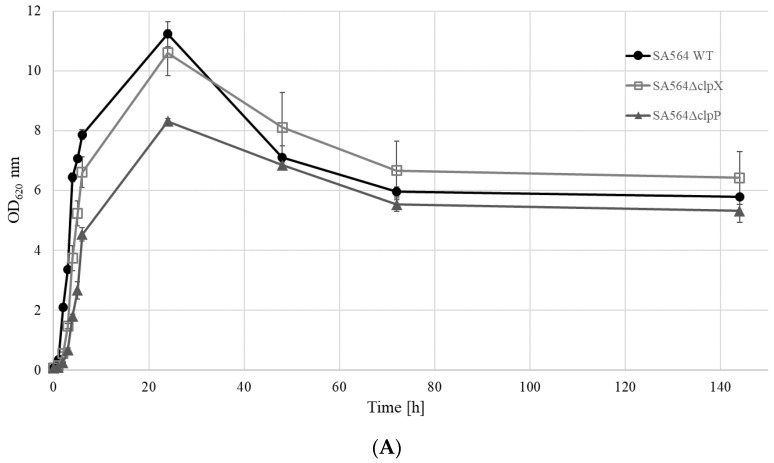
SED and TSST-1 protein levels are reduced in supernatants derived from cells lacking ClpXP activity. (**A**) Growth curves of the Sa564 wild-type and mutant strains in TSB at 37 °C. (•) SA564 wild-type, (□) SA564Δ*clpX*, and (▲) SA564Δ*clpP*. Average values and standard deviations of three independent biological replicates (*n* = 3) are shown. (**B**) Normalized SED production of the Sa564 wild-type and mutant strains. Dark-gray filled bars—SA564 wild-type; white bars—SA564Δ*clpX,* and light-gray filled bars—SA564Δ*clpP*. Average values and standard deviations of three independent biological replicates (*n* = 3) are shown. (**C**) Normalized TSST production of the Sa564 wild-type and mutant strains. Dark-gray filled bars—SA564 wild-type, white bars—SA564Δ*clpX*, and light-gray filled bars—SA564Δ*clpP*. Average values and standard deviations of two independent biological replicates (*n* = 2) are shown.

**Figure 3 toxins-12-00553-f003:**
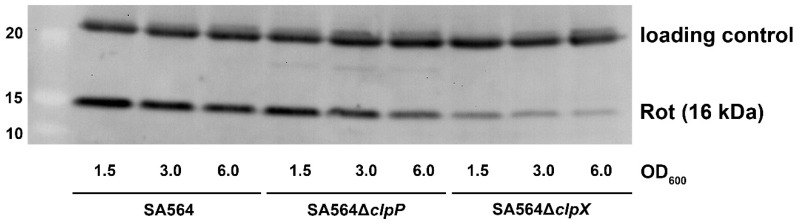
Diminished levels of the Rot repressor in cells with decreased transcription of the *sed, sec*, and *tst* genes. The cellular levels of the Rot transcriptional regulator were determined in the samples of SA564 wild-type, SA564∆*clpP*, and SA564∆*clpX* cells derived from the same cultures, as used for the RNA extraction (see the Northern blotting analysis presented in Figure 1). Accordingly, proteins were extracted from cells in late-exponential (OD_600_ = 1.5), early post-exponential growth phase (OD_600_ = 3.0), or late post-exponential growth phase (OD_600_ = 6.0), and the extracted proteins were separated by SDS–PAGE, blotted onto a PVDF membrane, and probed with anti-Rot antibody. The protein ladder is shown to the left. The upper band resulting from non-specific binding of the Rot antibody to an unknown cellular protein visualizes equal loading.

**Figure 4 toxins-12-00553-f004:**
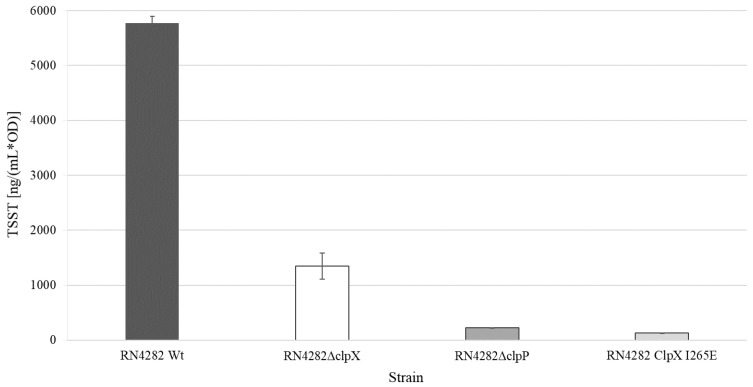
TSST-1 levels are reduced in supernatants derived from RN4282 cells lacking ClpXP activity. Normalized TSST production of RN4282 wild type and mutant strains. Dark-gray filled bar—RN4282 wild-type; white bar—RN4282Δ*clpX*; gray filled bar—RN4282Δ*clpP*, and light-gray bar—RN4282ClpX_I268E_. Average values of three technical replicates from one independent biological replicate (*n* = 1) are shown.

**Table 1 toxins-12-00553-t001:** Virulence regulator genes significantly differentially expressed * between post-exponential RN4282 wild-type cells and RN4282 cells devoid of ClpXP activity.

Gene	Fold Changes Wild-Type/Mutant	*p* _adjusted_
sarU	5.8	5.91 × 10^−14^
sarX	4.1	4.06 × 10^−13^
sarS	3.1	2.14 × 10^−7^
srrA	2.5	1.28 × 10^−4^
agrA	2.3	3.71 × 10^−6^
mgrA	2.1	3.17 × 10^−4^
RNAIII	2.0	8.80 × 10^−3^
sarT	0.5	1.57 × 10^−2^

* Genes were considered to be differentially expressed if they showed a ≥2-fold change in expression with *p*_adj_ < 0.05.

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
