# Peer review of "A Functional ClpXP Protease is Required for Induction of the Accessory Toxin Genes, tst, sed, and sec"

_toxins, 2020, doi:10.3390/toxins12090553_

Round 1
Reviewer 1 Report
The authors provide an interesting work on transcriptional regolators on the role of ClpXP protease in S. aureus. However, many issues are present. Among them, Northern and not qPCR in figure 1? probe lenght used for hybridisation? Level of RNA III are not correctly normalised. In line 153 you stated that "decreased levels of RNAIII in the mutants at OD600 = 6.0 (Figure 1)" were measured. In my opinion the rRNA in used to normalised RNAIII in Nortern blot do not support such assumption as decreased amount od total rRNA correspond to decresed RNA III. Please, you need to correct this, without this data the manuscript has serious methodological errors.Author Response
We would like to thank the reviewer for the fast response and valuable comments.
Please see below our comments to the raised points.
Figure 1: Northern blotting similar to RT-PCR is a standard method used for quantification of mRNA levels. RT-PCR, because of the reverse-transcriptase- and the PCR-amplification steps, has the risk of amplifying errors as well as signals. In contrast, Northern blotting allows direct visualization of the transcript with little fear of interference from whatever else is in the sample. The Northern blot, moreover reveals the size of the transcript and allows for observation of transcript processing or degradation. For normalization, a fixed amoun of RNA is used, and again the equal loading is visualized directly by the intensity of the rRNA bands. For these reasons, the authors find that Northern blotting is a more reliable method for quantifying transcription. In the present manuscript, the transcriptional changes detected using Northern blotting are confirmed using RNA-sequencing, as well as being reflected in the level of toxin produced.
Line 322-327: the lengths of the internal gene fragments used as templates in the labeling reactions have been added as requested by the reviewer.
Line 153: we see the point of the reviewer, and have rephrased this sentence accordingly to emphasize the important notion that the cellular level of Rot is decreased in SA564ΔclpX despite that the RNA level is not increased, Lines 153-156.

Reviewer 2 Report
In this study the authors outline the potential role of the ClpXP protease system in the regulation of three superantigen (SAg) genes. The manuscript does provide data that supports this hypothesis; however, I do have some concerns with the overall robustness of the data presented in this work. In many of the figure’s controls are missing or the wrong interpretation is drawn. Overall, it gives the impression of a sloppy study, which I don’t believe was the authors intention. I have really stressed where I think there are weaknesses and how to improve the robustness of the experiments.
Major Concerns
82-83 Can the authors explain why TSST-1 expression increases at OD 3.0 when glucose is added? Glucose is indeed a repressor of TSST-1 expression but there appears to be a discrepancy at this timepoint. Also, I cannot agree that glucose doesn’t affect SED expression as there is clearly inhibition of expression at the OD 3.0 when glucose is present. I am concerned the observations made in the results don’t align with the data presented and I think this reflects an issue with using northern blot analysis to make judgements on transcription. I am not insisting this be done, but I really think you should switch this analysis to a quantitative approach such as qRT-PCR or a transcriptional reporter plasmid.
While the data in figure 2c is quite clear, it is really frustrating the authors did not perform 3 biological replicates so stats could be performed on the data. Please do so for 2c and perform the stats on 2b as the data is already there.
Please provide the SDS-PAGE that accompanies the blot in figure 3, this is a critical control for loading bias. Also please provide some indication of the band size on the blot.
The lack of complements concerns me. Plasmids expressing ClpP and ClpX in trans in the null mutants would really add robustness to the data presented in figures 2 and 4. Please do not forget the empty vector controls.
Were all the mutants in figure 4 grown under erythromycin selection? If so, how are you controlling for stress responses induced by the antibiotic? This also raises concerns for me with the validity of the RNA-seq data on this mutant.
Where is the ClpP RNA-seq data?
Minor Concerns
Please define exactly what you mean for early post-exponential phase in terms of OD (i.e. OD 3.0) somewhere in the results text like was done for the late post-exponential phase.
Can you please state the concentration of glucose used and what carbon source was used as an alternative in the media not containing glucose in the data shown in figure 1.
I am unclear why the authors did not include SEC in the protein analysis.
Why are there 10 blank pages in the supplemental data. Is something missing?
Line 247 – of those strains listed that do not express SarU, to my knowledge non of them encode SED or TSST-1 so I am not so sure you can dismiss it so easily.
Author Response
We would like to thank the reviewer for the fast response and many valuable comments.
Please see below our comments to the raised points.
Author reply:
Specific comments
Line 82-83: The Northern blotting method and the experimental settings used to examine transcription of the tst gene in the presence of glucose was adopted from reference 34. We repeatedly observed glucose repression of tst transcription only at the latter time-point (OD 6). Glucose was added at OD=1.7, and we speculate that at OD 3, cells were exposed to glucose for too short a period to induce glucose repression.
Line 87-88: regarding sed-transcription in the presence of glucose, we see the point of the reviewer, and have changed the text accordingly.
General comment: Northern blotting similar to RT-PCR is a standard method used for quantification of mRNA levels. RT-PCR, because of the reverse-transcriptase- and the PCR-amplification steps, has the risk of amplifying errors as well as signals. In contrast, Northern blotting allows direct visualization of the transcript with little fear of interference from whatever else is in the sample. The Northern blot, moreover, reveals the size of the transcript, and allows for observation of transcript processing or degradation. For these reasons, the authors find that Northern blotting is a more reliable method for quantifying transcription. In the present manuscript, the transcriptional changes detected using Northern blotting are confirmed using RNA-sequencing, as well as being reflected in the level of toxin produced.
While the data in figure 2c is quite clear, it is really frustrating the authors did not perform 3 biological replicates so stats could be performed on the data. Please do so for 2c and perform the stats on 2b as the data is already there.
Author reply:
The statistical analyses of the different data sets (Fig 2B: SED in Sa564 strains, Fig 2C: TSST in Sa564 strains and Fig 4: TSST in RN4282 strains) have been performed in exactly the same way. Average values and standard deviations have been calculated of the different numbers of biological or technical replicates and two-tailed t-test have been used to evaluate the level of significance. In addition, the data was normalized against OD-values in order to take into account the differences in growth rate as toxin production mostly is correlated to growth. This is the data presented in each respective graph. For some results, the error bars are very small and thus not readily visible (eg for clpP-mutant in Fig 2C and RN4282 clpP-mutant, RN4282 ClpX I265E in Fig 4) in the graphs. We have proof read the material and methods section to verify that this is clearly and correctly explained.
The reason for not including a third biological replicate is of a technical aspect, as a new batch of antibodies had to be used for those samples. The age of antibodies have an impact on the signal intensity in the ELISA analysis and to combine data from different batches of antibodies is not optimal. The analysis of the samples of the third biological replicate resulted in the same expression pattern and differences among the compared strains but the absolute amounts of toxin concentrations differed and we therefore decided to not include this set of data.
Indeed, we agree with the reviewer that it is unfortunate that we could not be consequent in the number of replicates included and presented, as this for sure would have been the preferred option. We are, however, fully confident that the data presented, even though based on two replicates, represents the true picture of the difference in TSST toxin production between the wildtype and mutant strains as the diminished TSST-1 production in the two mutants also followed the trend observed in the Northern blot analysis.
Please provide the SDS-PAGE that accompanies the blot in figure 3, this is a critical control for loading bias. Also please provide some indication of the band size on the blot.
Author reply
Figure 3 has been revised as suggested by the reviewer, and now shows a loading control and the protein ladder used to verify that the Rot antibody recognizes a band of the expected size (16 kDa). The figure legend has been changed accordingly.
The lack of complements concerns me. Plasmids expressing ClpP and ClpX in trans in the null mutants would really add robustness to the data presented in figures 2 and 4. Please do not forget the empty vector controls.
Complemented strains have not been included as whole genome
Author reply:
The robustness of the SA564 data (Fig. 2) comes from whole genome sequencing, showing that the in-frame deletion in clpX is the only genetic difference between SA564 wild-type and SA564ΔclpX*, ruling out that other undesired genetic changes are causing the downregulation. This important information has now been added in line, 285-287. In Fig. 4, we show that introducing a single amino acid change in RN4282 ClpX (ClpXI265E) similarly to deletion of clpX or clpP dramatically reduce expression of TSST-1. In our opinion, this is a very robust way of showing that ClpXP proteolytic complex is causing the downshift in TSST-1 expression.
* Bæk KT, Bowman L, Millership C, et al. The Cell Wall Polymer Lipoteichoic Acid Becomes Nonessential in Staphylococcus aureus Cells Lacking the ClpX Chaperone. mBio. 2016;7(4):e01228-16. Published 2016 Aug 9. doi:10.1128/mBio.01228-16
Were all the mutants in figure 4 grown under erythromycin selection? If so, how are you controlling for stress responses induced by the antibiotic? This also raises concerns for me with the validity of the RNA-seq data on this mutant.
Author reply:
All strains were grown in the absence of erythromycin in Figure 4 and for the RNAseq analysis (as specified in the experimental section lines 292-2987+305-306 + 368-307. Erythromycin was only supplemented in the TSB medium during resuscitation of the cells after long-term storage as described in lines 287-289).
Where is the ClpP RNA-seq data?
Author reply:
The result of the RNAseq data comparing transcription of the RN4282 wild-type and the RN4282 expressing the ClpXI265E variant is shown in Table S1 (all genes significantly differentially expressed between post-exponential RN4282 wild-type cells and RN4282 cells devoid of ClpXP activity) and in Table 1 showing virulence gene regulators with changed expression.
Minor Concerns
Please define exactly what you mean for early post-exponential phase in terms of OD (i.e. OD 3.0) somewhere in the results text like was done for the late post-exponential phase.
Thanks for pointing this out. The definitions are now stated in lines 77-80.
Can you please state the concentration of glucose used and what carbon source was used as an alternative in the media not containing glucose in the data shown in figure 1.
This information is now stated in line 86. Tryptic Soy broth provide the carbon source in the culture without glucose added.
I am unclear why the authors did not include SEC in the protein analysis.’
It would indeed have been interesting to examine the SEC levels as well, however, as we did not have access to the SEC antibodies in house, we were unfortunately not able to perform these analyses during the time frame of this research.
Why are there 10 blank pages in the supplemental data. Is something missing?
Thank you for pointing out this mistake. The blank pages have been removed from the revised Table S1.
Line 247 – of those strains listed that do not express SarU, to my knowledge non of them encode SED or TSST-1 so I am not so sure you can dismiss it so easily.
Agreed.

Round 2
Reviewer 1 Report
dear Authors,
I agree with the use of Northern blot in the analyses of RNA stability as well as in processing and decay. However in the manuscript you provide no functional roles in such processes have been addressed. Thus, in my opinion the qPCRs are mandatory for this measurement.
Reviewer 2 Report
Thank you for addressing all my concerns. The manuscript is a lot clearer now and presentation is greatly improved. Its a very nice study, well done.